# Factors influencing SARS-CoV-2 IgG test sensitivity: A Bayesian analysis of seroconversion and seroreversion by time since infection, test, age and disease severity

Toon Braeye [1,2]*, Steven Abrams[2,3], Niel Hens[2,4]

**1** Department of Epidemiology and Public Health, Sciensano, Brussels, Belgium, **2** Interuniversity Institute for Biostatistics and statistical Bioinformatics (I-Biostat), Data Science Institute (DSI), UHasselt, Hasselt, Belgium, **3** Global Health Institute, Family Medicine and Population Health (FAMPOP), University of Antwerp, Antwerp, Belgium, **4** Centre for Health Economics Research and Modelling of Infectious Diseases (CHERMID), Vaccine & Infectious Disease Institute (VAXINFECTIO), University of Antwerp, Antwerp, Belgium

* toon.braeye@sciensano.be

## Abstract

### Background

Antibody testing is commonly used to assess past exposure to pathogens, but the interpretation is complex. We quantified test-specific SARS-CoV-2 seroconversion and seroreversion by time since PCR-confirmed infection, age and disease severity.

### Methods

We combined Belgian data from laboratory SARS-CoV-2 testing, prescriptions, contact tracing and hospital surveillance collected between March 2020 and June 2021 with data from published longitudinal studies on Wantai and EuroImmun IgG serological tests.

We used a hierarchical Bayesian model to estimate time-varying sensitivity of serological tests following PCR-confirmed infection. The model employed a scaled Weibull-bi-exponential distribution. We accounted for disease severity (distinguishing between asymptomatic, symptomatic, and hospitalized cases), age (i.e., age groups 18–49, 50–64, and 65–74 years) and serological test used.

### Results

We included 44,262 serological test results: 10,864 obtained from published studies, 33,398 from Belgian laboratories. Seroconversion occurred during the six weeks following a PCR-confirmed infection. Age, disease severity and the test used strongly influenced seroconversion rates and the rate of the subsequent seroreversion. For the EuroImmun test, 82% (95% Credible Interval (CrI): 80%−84%) of symptomatic

which permits unrestricted use, distribution, and reproduction in any medium, provided the original author and source are credited.

**Data availability statement:** All relevant aggregated data from the published studies included in our analysis are available in S1 Table. Highly granular, aggregated data from the Belgian laboratories (number of samples and seropositivity, stratified by time since infection, age group, sex, and clinical severity) are provided in the Supporting information files (S2 and S3 Tables). The raw, individual-level Belgian patient data used in this study contains sensitive, potentially identifying information and is subject to legal and ethical restrictions under Belgian and EU privacy regulations (GDPR). This data was sourced by linking multiple national health databases within a secure pseudonymized environment (LINK-VACC). The minimal aggregated dataset required to fully replicate the hierarchical Bayesian model (which includes the random effect for the reporting laboratory) is also restricted and cannot be made publicly available. Publishing the data stratified to this final level (by laboratory, age, severity, and time(in weeks)) would result in numerous small cell counts, creating risk of re-identification. Access to this restricted-use dataset for verification and replication purposes may be requested by researchers through Belgium's health data agency, https://www.hda.belgium.be/en.

**Funding:** The author(s) received no specific funding for this work.

**Competing interests:** The authors have declared that no competing interests exist.

individuals in the youngest age group seroconverted, compared to 95% (CrI: 95%−96%) for the Wantai test. Seroconversion was associated with hospitalization, (OR = 8.17 (CrI: 5.56–13.72), compared to asymptomatic infection) and older age (OR = 1.65 (CrI: 1.41–1.97), compared to 18–49 year-olds). Slower seroreversion was associated with older age, hospitalization and the Wantai test. At 50 weeks, seropositivity among symptomatic 18–49 year-olds was 64% (CrI: 58%−70%) for the EuroImmun test and 95% (CrI: 94%−96%) for the Wantai test.

## Conclusion

These findings highlight the need for test-specific, time-varying sensitivity adjustments in seroprevalence studies. Such adjustments are crucial for translating seroprevalence results to cumulative incidence estimates.

---

## 1. Introduction

Starting in 2020, numerous seroprevalence studies were conducted to understand the extent of past exposure to SARS-CoV-2 within different populations [1]. These studies used serological tests (immunoassays) capable of directly or indirectly detecting antibodies. These antibodies typically target one of the four structural proteins of the SARS-CoV-2 virus: spike (S), membrane (M), envelope (E), or nucleocapsid (N) protein. The S protein, which is further divided into the N-terminal domain (NTD) and the receptor-binding domain (RBD), along with the N protein, are the primary immunogens and are typically the targets of immunoassays [2]. Assays determine the amount of antibodies and manufacturer-suggested threshold values can be used to translate quantitative to qualitative results. Qualitative results are then typically reported as the proportion of positive samples among all samples. The interpretation of such results, however, is not straightforward as immunoassays are imperfect. Qualitative results are associated with a proportion of false negatives (sensitivity below 100%) and false positives (specificity below 100%) [3]. In addition, the main research interest is often not solely in the presence of antibodies. The main objective of seroprevalence studies, next to objectives regarding susceptibility, concerns the proportion of persons previously infected: the cumulative infection rate. To accurately estimate the cumulative infection rate, we must define sensitivity not merely as the detection of antibody titers above a threshold, but as the ability to detect a previous infection. This form of sensitivity is a dynamic metric that depends on the specific test, antibody kinetics over time, and individual characteristics [3].

Quantifying this dynamic sensitivity is difficult as it is affected by different factors. We briefly introduce the three main factors: (A) how infections are diagnosed, (B) which persons are included in the cohort under follow-up and (C) how time since infection is included in the analysis. With respect to (A): studies typically use either a 'gold standard' immunoassay or a combination of immunoassays, neutralization assays or RT-PCR tests to determine 'true' positivity of an individual [4]. The shortcoming of any of these options typically boils down to the gold standard's own

sensitivity and specificity. (B) Patients included in longitudinal studies were frequently hospitalized. Hospitalization is typically associated with high antibody titers, not reflective of titers in asymptomatic patients [5–7]. Meta-reviews reported a high risk of patient selection bias in 97−98% of assessments [4,8]. (C) After an initial increase during 3–12 weeks for SARS-CoV-2 [8], antibody titers will decrease again over time, while antibody avidity and affinity might increase. The decline is not monophasic. An initial strong decrease is followed by a plateau [9,10]. This decrease shows antigen-specific patterns. Antibodies associated with S1 are more durable than those associated with the N-protein [2,11–16], though the magnitude of this difference may vary by assay type and vaccination status [17]. Furthermore, the decrease is linked to patient selection. The half-time of IgG S-protein titers associated with asymptomatic cases is less than half of that of mild cases and has been estimated at 55 days [18,19] or even shorter (i.e., 36 days as estimated by Ibarrondo et al. [20]). No detectable neutralizing activity was found in 50% of asymptomatic infections one year after infection [10]. The role of sex and age is less clear [21]. More stable antibody levels have been reported for females [22]. Studies have reported higher initial titer concentrations in older age groups. Higher initial titers have been linked to extended seropositivity [7,14,22–24]. Typical methods to adjust for the imperfect performance of serological tests, such as a Rogan-Gladen type estimator [25], can be extended to include time-varying sensitivity estimates. However, except for some modelling studies, seroprevalence studies typically do not correct for antibody waning [26].

We aimed to estimate time-varying, test-specific sensitivity in relation to time since infection. We also quantified the effects of age and clinical severity on this sensitivity. To estimate sensitivity, we included data from Belgian laboratories and international data from the literature. Accurate estimates for the time-varying and test-specific sensitivity are necessary for the translation of population-level seroprevalence results to (cumulative) incidence estimates.

## 2. Methods

Data on cases were collected from Belgian laboratories and published studies. Belgian laboratory data contributed large sample sizes, but lacked details on test type. Published longitudinal studies provided small, well-characterized cohorts with explicit test identification. Analogous to meta-analytic methods, we synthesized evidence across data sources to include all information and increase statistical precision. We used a Bayesian hierarchical model with a binomial likelihood to fit the number of positive serological tests out of all serological tests at week $t$ after PCR-confirmation of SARS-CoV-2 infection.

### 2.1. Laboratory data

**2.1.1. Data source.** Reporting of laboratory results to a centralized database was mandatory during the COVID-19 pandemic in Belgium. While SARS-CoV-2 IgG testing was not required in any situation, a large number of IgG tests were performed and reported to the centralized database. This database could be linked to other databases. For this study, linkage was made with the contact tracing database (results from case interviews, including self-reported symptoms), the vaccinnet+ database (COVID-19 vaccination registry), the DEMOBEL database (age, sex, place of residence) and the clinical hospital surveillance. Linkage was based on a pseudonymized identifier obtained from the encryption of an individual's national registry number and performed in LINK-VACC, a secure pseudonymized environment hosted by Healthdata.be at the Belgian Institute for Public Health.

**2.1.2. Data structure.** We grouped test results by severity, age, test/laboratory and number of weeks since PCR-confirmed infection. We considered three age groups: 18–49 years, 50–64 years and 65–74 years and three different categories of clinical severity: self-reported asymptomatic, self-reported symptomatic or notified hospitalized. Symptoms could be reported either before diagnosis (e.g., during the consultation as reported on the prescription form) or during contact tracing (when cases were interviewed about their contacts). If the case reported symptoms at any point, the case was classified as symptomatic. If, upon investigation, the case reported no symptoms, it was classified as asymptomatic. If this information was missing, the records were excluded. Additionally, if a hospitalization for COVID-19 was reported

from 7 days before to 60 days after the first positive PCR test through the clinical hospital survey, the case was classified as hospitalized [27]. We explored sex as an additional covariate, however, initial analysis did not indicate a significant association between sex and test positivity. Descriptive statistics for age, severity and sex are included in S2 and S3 Tables. Unfortunately, the specific serological test and cut-offs used were not reported by the laboratory. The reporting laboratory was therefore used as a substitute for test and included as a random variable in the model.

**2.1.3. Exclusion criteria.** We restricted analysis to PCR-positive cases from 2020. This avoided confounding from new variants emerging in 2021, which altered epidemiology and immunological responses. We excluded all IgG tests that followed any subsequent PCR test after the first positive test as PCR-testing could be indicative of additional exposure to SARS-CoV-2. Belgium started the mass roll-out of the vaccination campaign in January 2021, prioritizing nursing home residents and healthcare workers. The further roll-out was age-based, with persons below the age of 65 years invited from April 2021 onwards. For individuals in our study cohort who subsequently received COVID-19 vaccination, we excluded any IgG test results obtained after their vaccination date. No IgG test results obtained after June 2021 were included.

## 2.2. Literature data

**2.2.1. Data source.** We included two tests, chosen because of their use in two repeated cross-sectional seroprevalence studies in Belgium, a semi-quantitative test (EuroImmun), targeting the S1 protein and a semi-quantitative test (Wantai) targeting the RBD. We included all studies listed in the systematic review by Owusu-Boaitey et al. [3] in our statistical analysis. For the Wantai test, we also included the findings from the study by Hønge et al. [28], originally excluded by Owusu-Boaitey et al. because the study cohort, consisting exclusively of healthy blood donors, was not considered representative. Since our model allowed us to differentiate by disease severity, we included the study by Hønge et al. [28].

**2.2.2. Data structure.** The data from scientific literature was included in the model in a similar way to the data obtained from Belgian laboratories. As opposed to the data from the Belgian laboratories, the specific test is known and included in the model as a level for the random effect $u_{test/lab_l}$. For each included paper, we obtained the test, the number of (positive) samples by weeks since infection, the proportions of asymptomatic, symptomatic and hospitalized persons in the cohort and the proportion in each age group. Whenever specifics on severity and age were not available, we included the following default distributions. For clinical severity: 50% asymptomatic, 45% symptomatic and 5% hospitalized (severe) as this was reported by Takahashi et al. [29] to be the average over serosurveys. For age: we either included the default for Germany, if the study was performed in Germany: 18–49 (45%), 50–64 (35%) and 65–74 (20%) based on Neuhauser et al. [30] or the default for Denmark: 18–49 (64%), 50–64 (26%) and 65–74 (10%) based on Pires et al. [31]. For studies and regions outside of Denmark and Germany for which we could not obtain a specific age distribution, we used as default a distribution in between the German and Danish distributions: 18–49 (55%), 50–64 (30%) and 65–74 (15%). Details by study and weeks since infection are provided in S1 Table.

## 2.3. Model structure

**2.3.1. Seroconversion and seroreversion.** We modeled two immunological processes: seroconversion and seroreversion. Seroconversion is the process of reaching a detectable level of antibodies after infection. Seroreversion is the process of losing that detectable level after having first attained it. We represented these as time-to-event distributions: a Weibull distribution for time to seroconversion ($T_c$) and a bi-exponential distribution for time to seroreversion ($T_r$). The Weibull distribution flexibly accommodates varying seroconversion rates. The bi-exponential distribution captures the biphasic nature of antibody decay—an initial rapid decline followed by slower waning. By combining these distributions, we calculated the proportion of individuals remaining seropositive at any given time since infection ($S_t$), with parameters varying by age, disease severity, and test type. We used Bayesian inference to estimate model parameters from observed test results, allowing us to quantify uncertainty in seroconversion and seroreversion estimates.

 

**2.3.2. Bayesian model.** In mathematical terms: the random variable $S_t$, with discrete probability density function $h(S_t)$, represents the proportion of seropositive cases at week $t$ after infection. We account for the distribution of seroconversion times by summing over all possible weeks of conversion up to week $t$. For individuals who seroconverted at any week $t_c$, we evaluate the probability they remain seropositive (have not undergone seroreversion) by week t using $[1 - Z_r(t - t_c)]$. In equation 1, $g(t_c)$ is the discretized probability density function for $T_c$, $Prop$ represents the proportion of cases that will eventually seroconvert, and $Z_r$ denotes the cumulative probability of seroreversion evaluated at discrete weekly time points.

$$h(S_t) = \sum_{t_c=1}^{t} g(t_c) * Prop * [1 - Z_r(t - t_c)] \tag{1}$$

$Z_r$ combines fast ($\lambda_{r.fast}$) and slow ($\lambda_{r.slow}$) exponential decay processes with mixing proportion $\alpha_r$ as shown in equation 2.

$$Z_r(t - t_c) = 1 - (\alpha_r * exp(-\lambda_{r.fast} * (t - t_c)) + (1 - \alpha_r) * exp(-\lambda_{r.slow} * (t - t_c))) \tag{2}$$

Six parameters need to be estimated: a scale and shape parameter for $T_c$ ($scale_c$, $shape_c$), two exponential parameters for $T_r$ ($\lambda_{r.fast}$, $\lambda_{r.slow}$), the proportion of cases that undergo seroconversion and $\alpha_r$ which determines the mixture between the slow and fast exponential decay. The parameters for the time-to-event distributions ($scale_c$, $shape_c$, $\lambda_{r.fast}$, $\lambda_{r.slow}$) are shared by the different groups. The linear predictors for $Prop$ and $\alpha_r$ are shown in equation 3 and 4.

$$logit(Prop) = \sum_{i=1}^{2} \beta_{pc,1i} * severity_i + \sum_{j=1}^{2} \beta_{pc,2j} * age_j + u_{pc.test/lab_l} \tag{3}$$

$$logit(\alpha_r) = \sum_{i=1}^{2} \beta_{r,1i} * severity_i + \sum_{j=1}^{2} \beta_{r,2j} * age_j + u_{r.test/lab_l} \tag{4}$$

The Odds Ratios (OR) obtained from these coefficients quantify the relative odds of seroconversion (equation 3) or fast decay (equation 4) between groups. To improve identifiability of the bi-exponential mixture components, we anchored $\alpha_r$ at −6 (on the logit scale) for the reference group of hospitalized patients aged 65–74 years. This choice was informed by exploratory analysis showing minimal seroreversion in this group, effectively defining them as exhibiting predominantly slow antibody decay. Setting $\alpha_r$ = –6 corresponds to approximately 0.25% weight on the fast decay component for this reference group, allowing other age and severity groups to be estimated relative to this anchor point. $u_{test/lab_l}$ represent the random effect for tests and laboratories.

We opted for Bayesian inference since it allows for simultaneous estimation of global parameters (shape of seroconversion and seroreversion curves), group-specific parameters (effects of age and severity), and test/laboratory-specific random effects, with appropriate propagation of uncertainty across all levels. In addition, our model includes complex non-linear functions (Weibull and bi-exponential distributions), which would make likelihood-based inference challenging.

**2.3.3. Model selection.** We evaluated multiple approaches for modelling seroreversion kinetics, including linear decay functions and various time-to-event distributions such as Weibull. After comparative analysis, we selected a bi-exponential distribution. This model optimally captured the biphasic nature of antibody dynamics: an initial rapid decline followed by a slower, plateau-like phase. While $\alpha_r$ was allowed to vary by severity, age group, and test, we determined through model selection that the decay rate parameters $\lambda_{r.fast}$ and $\lambda_{r.slow}$ could be treated as global parameters without significant loss

of fit. Multiple model structures were evaluated using Bayesian Information Criterion (BIC) and convergence diagnostics, including interaction terms.

**2.3.4. Priors, model fit and convergence.** We used diffuse normal priors (mean = 0, SD = 100) for model coefficients. Sensitivity analyses with more informative priors (SD = 5) produced nearly identical parameter estimates. The factors for test (or lab) were included as random effects with mean zero and a gamma distributed prior for the standard deviation. Markov Chain Monte Carlo (MCMC) sampling was performed using the R package *nimble*. We used three MCMC chains with 18,000 iterations each and a burn-in of 6,000 iterations to perform posterior inference. Convergence of the different chains was checked using trace plots and the Gelman-Rubin statistic (S1 Fig). All credible intervals (CrI, the Bayesian analogue of confidence intervals) are 95% unless otherwise stated.

## 3. Results

### 3.1. Numbers included

In the centralized database there were 472,223 persons with a positive PCR test in 2020 in Belgium in the age group from 18 years to 74 years. Of these persons 15% (N = 70,951) had IgG tests before July 2021. The total number of IgG tests associated with these persons was 93,127. We excluded, in order of exclusion criterion: 6,856 IgG tests following vaccination, 39,622 tests preceding the first positive PCR test or following any PCR test after the first positive PCR test. For 13,251 IgG tests, data on the severity of infection was missing. A flowchart of the exclusion is presented as S2 Fig. Finally, we included 33,398 IgG tests from 30,002 persons reported by 84 laboratories. The laboratory with the most records accounted for 14% of all IgG tests.

Of these tests 28% were taken from individuals with asymptomatic infection, 67% from individuals with symptomatic infection and 5% from hospitalized individuals. The division by age group was 51% (18–49 years), 35% (50–64 years) and 14% (65–74 years). 16% of tests were collected in the first 4 weeks, 42% in weeks 4–11, 40% in weeks 12–35 and 1% later. The numbers by weeks since first positive PCR test, clinical severity and age group are presented in Table 1.

### 3.2. Hierarchical Bayesian model

The proportion that eventually undergoes seroconversion was associated with test, disease severity and age. More specifically, the odds ratio of seroconversion were 2.25 (CrI: 1.99–2.58) for symptomatic and 8.17 (CrI: 5.56–13.72) for hospitalized individuals as compared to self-reported asymptomatic individuals. Higher seroconversion was associated

**Table 1. Number (and percentage) of included IgG tests following a positive SARS-CoV-2 PCR test in 2020, by weeks since positive PCR test, clinical severity and age group, Belgian laboratory data.**

| Severity | Weeks since positive PCR test | 18-49 years | 50-64 years | 65-74 years |
|---|---|---|---|---|
| Asymptomatic | 0-3 | 1,247 (3.7) | 687 (2.1) | 285 (0.9) |
| | 4-11 | 1,854 (5.6) | 1,139 (3.4) | 506 (1.5) |
| | 12-35 | 1,917 (5.7) | 1,083 (3.2) | 400 (1.2) |
| | 36+ | 145 (0.4) | 50 (0.1) | 8 (0) |
| Symptomatic | 0-3 | 1,430 (4.3) | 965 (2.9) | 421 (1.3) |
| | 4-11 | 5,125 (15.3) | 3,520 (10.5) | 1,430 (4.3) |
| | 12-35 | 5,063 (15.2) | 3,357 (10.1) | 1,082 (3.2) |
| | 36+ | 64 (0.2) | 33 (0.1) | 4 (0) |
| Hospitalized | 0-3 | 102 (0.3) | 135 (0.4) | 162 (0.5) |
| | 4-11 | 129 (0.4) | 288 (0.9) | 198 (0.6) |
| | 12-35 | 121 (0.4) | 266 (0.8) | 167 (0.5) |
| | 36+ | 5 (0) | 4 (0) | 6 (0) |

with older age: for 50–64 year-olds the OR was 1.29 (CrI: 1.13–1.45) and for 65–74 year-olds the OR was 1.65 (CrI: 1.41–1.97) compared to the youngest age group (18–49 years old) (Table 2). Higher values of $\alpha_r$ indicate greater weight on the fast decay component and thus faster antibody waning. Symptomatic infection, OR = 2.09 (CrI: 1.11–4.24), was associated with faster waning compared to asymptomatic infection, while older age groups showed substantially slower waning (Table 3).

Substantial variation in antibody dynamics was attributed to test/laboratory. The standard deviation of test-specific random effects for the mixture parameter $\alpha_r$ was 2.15 (CrI: 1.52–3.02). For proportion seroconverting, the SD was 0.55 (95% CrI: 0.41–0.72), corresponding to OR = 0.34 to OR = 2.98 across tests. For context, among symptomatic individuals aged 18–49 years, the predicted seroconversion at week 6 was 82% (CrI: 80%−84%) for EuroImmun and 95% (CrI: 95%−96%) for Wantai, while the proportion remaining seropositive at week 50 was 64% (CrI: 58%−70%) for EuroImmun and 95% (CrI: 94%−95%) for Wantai. The effect of the different tests/laboratories is further illustrated through the distributions of the random effects in S3 Fig.

**Table 2. Odds Ratios (OR) derived from the posterior distributions of the regression coefficients for the proportion of cases that eventually undergoes seroconversion after a positive PCR-test in 2020 (asymptomatic = self-reported asymptomatic, symptomatic = self-reported symptomatic, hospitalized = notified hospitalized, SD = standard deviation), Belgian laboratory data and data from published research.**

| Variable | OR | 95% CrI: Lower | 95% CrI: Upper |
|---|---|---|---|
| Age | | | |
| 18–49 (ref) | 1 | | |
| 50-64 | 1.29 | 1.13 | 1.45 |
| 65-74 | 1.65 | 1.41 | 1.97 |
| Severity | | | |
| Asymptomatic (ref) | 1 | | |
| Symptomatic | 2.25 | 1.99 | 2.58 |
| Hospitalized | 8.17 | 5.56 | 13.72 |
| Random intercept | SD | | |
| test/lab | 0.55 | 0.41 | 0.72 |

**Table 3. Odds Ratios (OR) derived from the posterior distributions of the regression coefficients for the odds of belonging to the fast decay group versus the slow decay group (asymptomatic = self-reported asymptomatic, symptomatic = self-reported symptomatic, hospitalized = notified hospitalized, SD = standard deviation), Belgian laboratory data and data from published research.**

| Variable | OR | 95% CrI: Lower | 95% CrI: Upper |
|---|---|---|---|
| Age | 1 | | |
| 18–49 (ref) | | | |
| 50-64 | 0.18 | 0.07 | 0.31 |
| 65-74 | 0.08 | 0.03 | 0.19 |
| Severity | | | |
| Asymptomatic (ref) | 1 | | |
| Symptomatic | 2.09 | 1.11 | 4.24 |
| Hospitalized | 0.29 | 0.10 | 0.88 |
| Random intercept | SD | | |
| test/lab | 2.15 | 1.52 | 3.02 |

By week 5, over 95% of those that will seroconvert had seroconverted. Globally, the slow decay rate was close to zero (0.0002, CrI: 0–0.0004). Fast decay was estimated at 0.039 (CrI: 0.027–0.056). The EuroImmun test was associated with more and faster seroreversion compared to the Wantai test. Differences were smaller after hospitalization and for older age groups. For example, in 65–74 year-olds after hospitalization 96% remained seropositive with EuroImmun compared to 98% with Wantai. We present the posterior distributions for seroconversion and seroreversion separately by clinical severity and age group for the EuroImmun and Wantai test in Fig 1. Seropositivity over time since PCR-confirmed infection is presented in Fig 2.

We present the model fit to both Belgium's laboratory data (Fig 3) and the data obtained from literature (Fig 4). Posterior predictive checks showed close alignment between observed and predicted proportions.

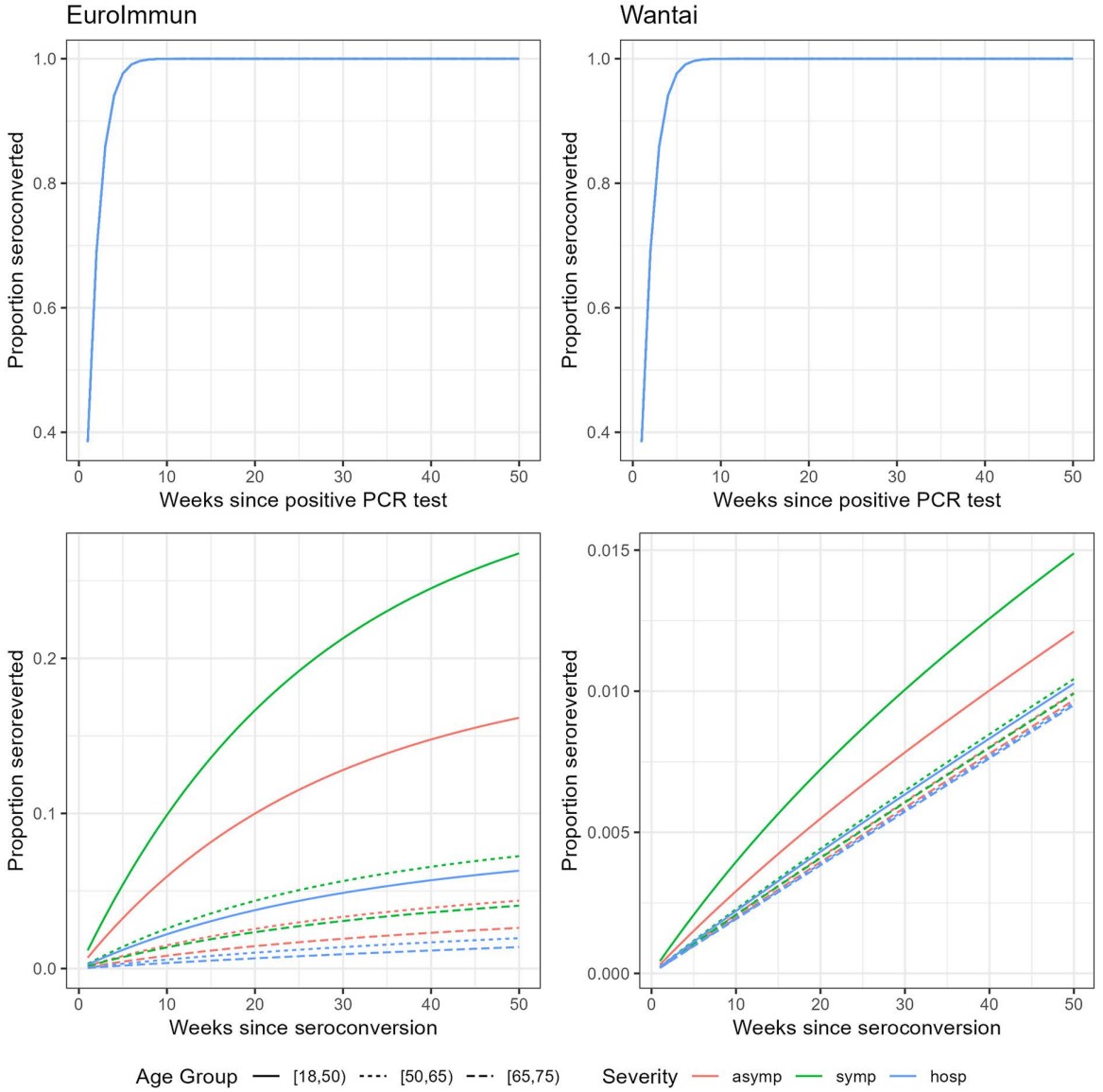

**Fig 1. Plots of the posterior Weibull distributions for seroconversion (upper) and seroreversion (lower) by weeks since positive PCR test, clinical severity and age group for the EuroImmun (left) and the Wantai (right) test, Belgian laboratory data and data from published research.**

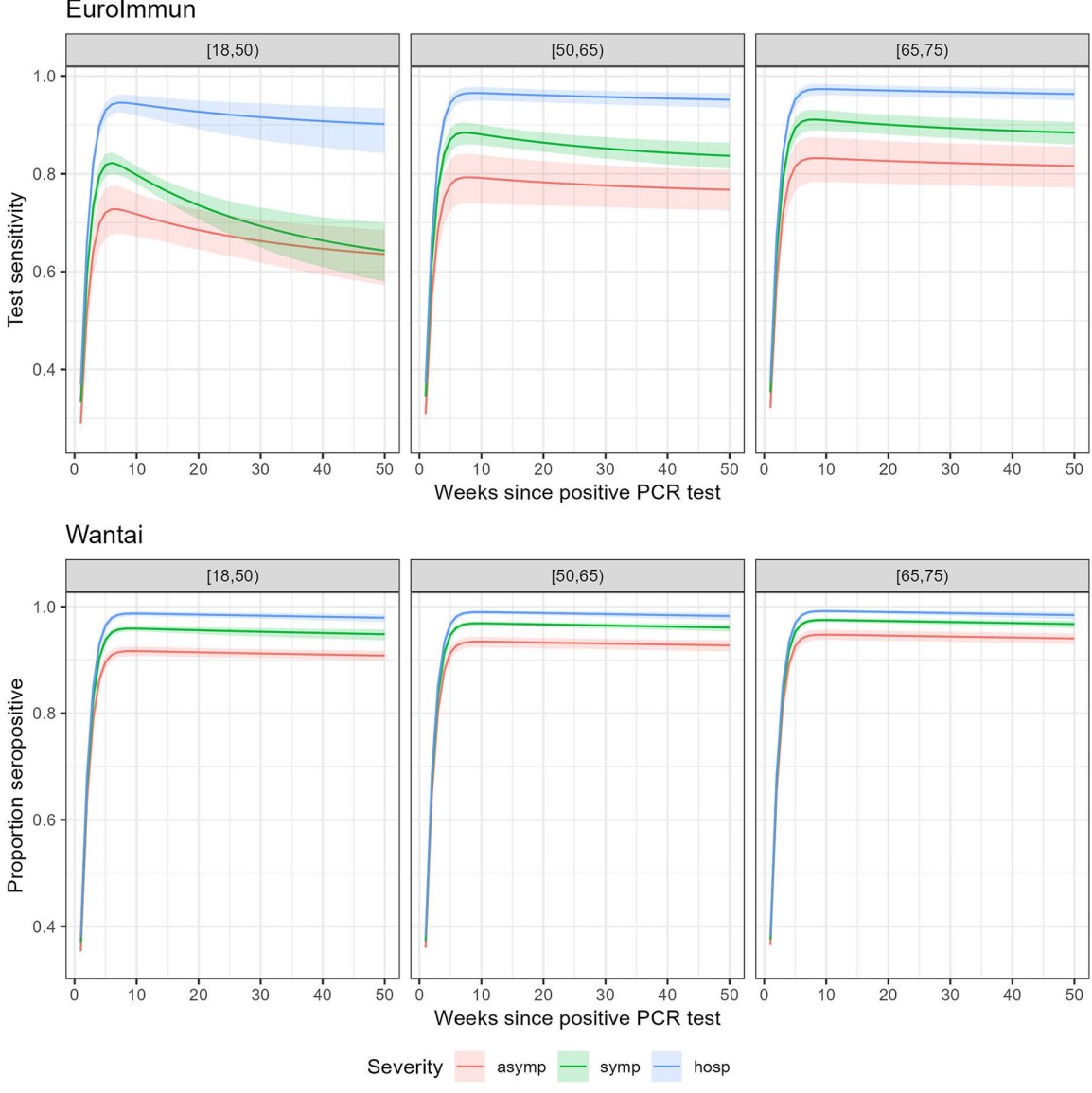

**Fig 2. Plots of the posterior scaled Weibull-Bi-exponential distribution for time-varying seropositivity for the EuroImmun (upper) and Wantai (lower) test by weeks since positive PCR test, clinical severity and age group, Belgian laboratory data and data from published research.**

## 4. Discussion

This study provides a comprehensive analysis of SARS-CoV-2 IgG test sensitivity by clinical severity, age, test used and time since infection. We quantified seropositivity using a hierarchical Bayesian model and included longitudinal data from published studies and Belgian laboratories. Our key findings demonstrate that seropositivity over time since SARS-CoV-2 infection is significantly influenced by three factors: the specific serological test used, the age of the individual, and the severity of the initial infection.

Adjusting qualitative seroprevalence results for test-specific sensitivity and specificity to estimate past exposure remains relatively rare. In a 2020 systematic review of global SARS-CoV-2 antibody seroprevalence, Bobrovitz et al.

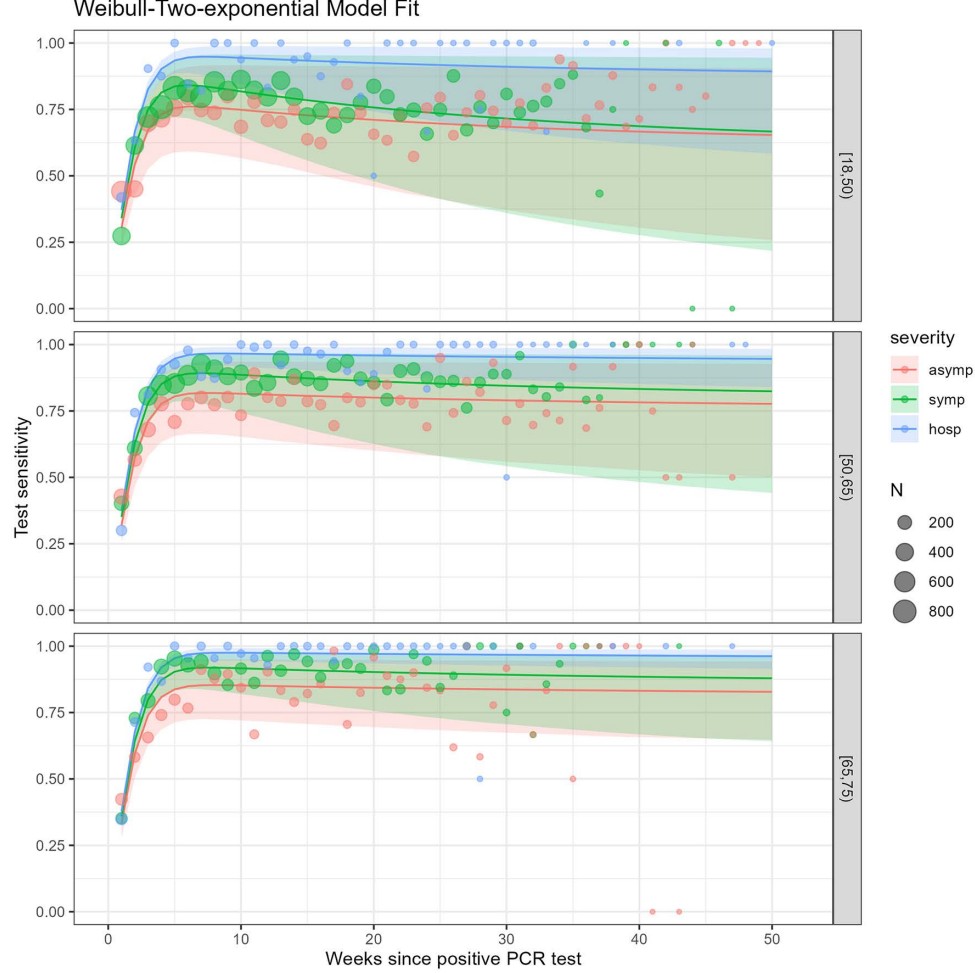

**Fig 3. Plots of the posterior scaled Weibull-Bi-exponential distribution for the time-varying seropositivity averaged over all laboratories (unweighted average) by weeks since positive PCR test, clinical severity and age group, Belgian laboratory data, IgG tests after a positive PCR test in 2020.**

reported that only 24% of studies provided sensitivity and specificity estimates, with an even smaller proportion adjusting their seroprevalence estimates accordingly [32]. The importance of these adjustments, however, has been demonstrated [17]. Studies using the same seroprevalence data, but different sensitivity estimates have reported considerably different infection fatality rates [33,34]. Their limited use may be attributed to the complexity of these metrics. Sensitivity depends not only on the test itself but also on the study cohort and time since infection. For example, for the EuroImmun serological test sensitivity estimates have been reported ranging from 94% to 53%. The manufacturer reported a sensitivity ≥10 days post symptom onset of 94.4%. Researchers in South Africa estimated sensitivity at 64.1% [6], Public Health England at 72%, 74.5% [35], 77.2% [36] and a study on Antarctic cruise passengers at 81% declining to 76% at 3 months and 53% at 1 year [37].

Previous Bayesian models combined data sources to estimate test sensitivity [38]. However, these frameworks did not account for time since infection or cohort characteristics. Given the high risk of patient selection bias associated with longitudinal sensitivity studies [4,8], the included cohort can't always be generalized to a wider population. The systematic review by Owusu-Boaitey et al. [3], for example, quantified test-specific sensitivity over time but could only include one

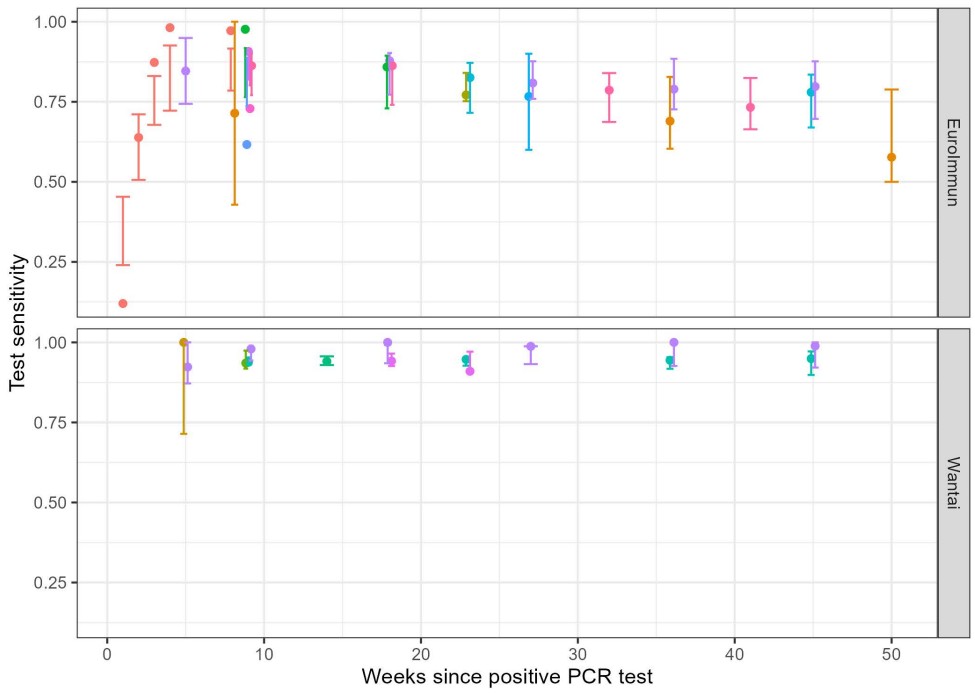

**Fig 4. Posterior Predictive Checks: The proportion positive reported (dot) and the posterior binomial credible interval (error bars) by study (color) and weeks since positive PCR test for the EuroImmun (upper) and Wantai (lower) test, data from published research.**

study [12] to estimate the sensitivity of the Wantai test beyond 5 months after symptom-onset. This study by Scheiblauer et al. included PCR-confirmed cases (N = 390) of which none were asymptomatic and a large proportion (15.8%) were hospitalized. Other studies on the Wantai test, such as a study on blood donors [28], could not be included because the cohort was not considered representative. In our study, we could include this population as we could account for clinical and demographic characteristics. In addition, we included studies too recent to be included by Owusu-Boaitey et al. [28,39]. Our study was limited to cohorts for which initial infection was documented through PCR-positivity. Studies that established a cohort by including persons with an initial positive IgG test [40] or with a specific set of symptoms [41] were not included.

Our study included two commonly used assays: EuroImmun and Wantai. EuroImmun detects antibodies to the S1 domain while Wantai targets the receptor-binding domain (RBD). In addition to differences in the targeted antigen, assay-specific characteristics – including which antibody classes are detected (total antibodies vs. IgG only), manufacturer cut-off thresholds, and epitope-specific binding characteristics – are expected to result in different time-varying sensitivity [17]. We observed a decrease in sensitivity over time for EuroImmun. For 18–49 year-olds with a symptomatic infection a drop of around 20 percentage points in seropositivity, corresponding to seroreversion in around 25% of those who seroconverted, occurred over a 50 week period. This sensitivity to time since infection partly explains the different sensitivity estimates for EuroImmun presented previously. In contrast, we observed no seroreversion for Wantai. Previous research reported 94.2%−98% seropositivity without seroreversion for at least 13–15 months after infection for Wantai [2,9,23,28,39].

While in our dataset Wantai was the most sensitive and durable serological test, it remains associated with a proportion of non-responders. Non-response, non-conversion or also sero-silence, is the absence of detectable antibodies at any time since infection. The proportion of sero-silence after a PCR-confirmed infection is estimated at 5.2–7.8% [42–44]. As with the previous findings this proportion also depends on the test used and the cohort characteristics.

Our other findings also largely agree with previously published longitudinal studies and reviews. The severity of infection has been associated with high and persistent antibody levels by several studies [45–50]. In addition, research reports lower sensitivity in young adults (aged younger than 40–50 years) [51–53], but the association in older age groups seems less clear with contradictory findings. Higher age has been mostly associated with slower waning [11,40,47,48,54,55] of the humoral immune response, with some studies reporting faster waning [22,49]. We found higher seropositivity in those aged over 50 as compared to those below, but the effect of age was smaller than the effect of severity of infection. We could not include persons beyond the age of 75 years. An age beyond which frailty and immunosenescence might impact sensitivity [56].

This study has several limitations. The process of seroreversion was included using a bi-exponential distribution. Some studies have opted for a Weibull distribution [57,58], others for a single exponential [59,60] or splines on the logit scale [61]. We did not extrapolate beyond the periods available within the data. Most of our data concerns the first weeks after PCR confirmation. As in many other studies, we set a positive PCR-test as reference. Laboratory confirmation by PCR, however, has its own time-varying sensitivity and specificity [62]. As no data were available on the test used by the reporting laboratory, we used the reported laboratory as level for a random effect. We did not fully explore sex as a cofactor as this information was missing in many of the included longitudinal studies. In our initial analysis, limited to the Belgian data, we found no evidence of heterogeneity in antibody persistence by sex. Some studies have come to a similar conclusion [63], while others claimed a more durable response in females [49]. The age groups in this analysis as well as the specific tests were chosen in preparation of future work on two cross-sectional seroprevalence studies.

In conclusion, our study provided a comprehensive framework for estimating time-varying, test-specific sensitivity in serological testing for SARS-CoV-2, accounting for the critical factors of disease severity, age, and time since infection. The results demonstrate that older age (50–74 years) and more severe infection lead to higher and more durable seropositivity. Our findings have several practical implications. First, test selection substantially affects sensitivity, particularly at longer intervals post-infection; the Wantai test has higher initial sensitivity which is maintained while EuroImmun showed considerable waning of its already lower sensitivity. Second, seroprevalence estimates from populations differing in age or severity composition are not directly comparable without adjustment. Third, sensitivity estimates derived from hospitalized cohorts—common in validation studies—will overestimate sensitivity in general populations with milder infections.

## Supporting information

**S1 Fig. Trace plots for the model's main parameters and their Gelman diagnostic statistic.**
(PDF)

**S2 Fig. Flowchart representing the inclusion criteria and associated numbers.**
(PDF)

**S3 Fig. Discrete approximation of the distribution of the random effects associated with test/lab for the random effect in proportion seroconverted (upper) and the $\alpha_r$ representing faster decay in seropositivity (lower).** The values associated with the EuroImmun and Wantai tests are annotated.
(TIFF)

**S1 Table. Data from published studies on seroconversion after PCR-confirmed infection for the EuroImmun (upper) and Wantai (lower) serological test.** Author refers to the first author, proportion by clinical severity (asymp, symp, hosp) and age group are presented per study. The notes describe the source of these proportions.
(DOCX)

**S2 Table. Number of observations included by sex, severity, age group and weeks since positive PCR test.** The percentage over the complete samples are presented within brackets. (Eg 10.7% of our sample were males, 18–49 years old, 4–11 weeks after symptomatic infection). Belgian laboratory data.
(DOCX)

**S3 Table. Percentage of seropositivity by sex, severity, age group and weeks since positive PCR test.** (Eg 79.6% of females, 18–49 years old, 12–35 weeks after symptomatic infection had a positive serological test). Belgian laboratory data.
(DOCX)

## Author contributions

**Conceptualization:** Toon Braeye, Steven Abrams, Niel Hens.

**Data curation:** Toon Braeye, Steven Abrams.

**Formal analysis:** Toon Braeye, Steven Abrams.

**Investigation:** Toon Braeye.

**Methodology:** Toon Braeye, Steven Abrams, Niel Hens.

**Supervision:** Toon Braeye, Niel Hens.

**Validation:** Toon Braeye.

**Writing – original draft:** Toon Braeye, Steven Abrams, Niel Hens.

**Writing – review & editing:** Toon Braeye, Steven Abrams, Niel Hens.

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
