## [Decision Letter · Decision Letter 0]

17 Sep 2025

Dear Dr. Braeye,

Thank you for submitting your manuscript to PLOS ONE. After careful consideration, we feel that it has merit but does not fully meet PLOS ONE’s publication criteria as it currently stands. Therefore, we invite you to submit a revised version of the manuscript that addresses the points raised during the review process.

https://journals.plos.org/plosone/s/submission-guidelines#loc-laboratory-protocols . Additionally, PLOS ONE offers an option for publishing peer-reviewed Lab Protocol articles, which describe protocols hosted on protocols.io. Read more information on sharing protocols at https://plos.org/protocols?utm_medium=editorial-email&utm_source=authorletters&utm_campaign=protocols .

We look forward to receiving your revised manuscript.

Kind regards,

Caio Bezerra Souto Maior, Ph.D.

Academic Editor

PLOS ONE

Journal Requirements:

Additional Editor Comments (if provided):

In addition to the reviewer's comments, if possible, I would suggest the authors discuss other works dealing with EuroImmun tests and/or Bayesian analysis for seroprevalence estimation, such as:

Lins et al. (2022). SerumCovid database: description and preliminary analysis of serological COVID-19 diagnosis in healthcare workers. PLoS ONE. https://doi.org/10.1371/journal.pone.0265016Maior et al. (2022). Seroprevalence of SARS-CoV-2 on health professionals via Bayesian estimation: a Brazilian case study before and after vaccines. Acta Tropica, 233, 106551. https://doi.org/10.1016/j.actatropica.2022.106551

Reviewers' comments:

Reviewer's Responses to Questions

**Comments to the Author**

1. Is the manuscript technically sound, and do the data support the conclusions?

Reviewer #1: Yes

Reviewer #2: Partly

2. Has the statistical analysis been performed appropriately and rigorously?

Reviewer #1: Yes

Reviewer #2: Yes

3. Have the authors made all data underlying the findings in their manuscript fully available?

Reviewer #1: No

Reviewer #2: No

4. Is the manuscript presented in an intelligible fashion and written in standard English?

Reviewer #1: Yes

Reviewer #2: Yes

Reviewer #1: The submitted paper presents a Bayesian framework for seroconversion and seroreversion analysis including time, age and severity factors, and including both literature and Belgium laboratory data. The methodology proposed appears robust and suggests possible applications in other contexts. The results suggest higher seroconversion for higher ages and as symptoms get more severe, while seroreversion was more common for lower ages and over the light-symptomatic group, almost not occurring over hospitalized patients.

Some studies include gender as a possible variable regarding seroconversion. From your Discussion section: “We opted not to explore sex because initial analysis did not associated sex with large differences in seropositivity after infection… With some claiming a more durable response in females [48].”. Your paper included references to studies showing differences (and also no differences) regarding gender. I believe including this factor in your analysis would make your study even more robust.

Please review your parenthesis usages along the article. In some paragraphs, such as in page 6 lines 116 to 119, there are too many, including parenthesis-inside-parenthesis, which makes the text a bit harder to read. I suggest that you consider changing some descriptions from parenthesis to commas in order to make the text cleaner, for example: “The random variable (with discrete probability density function (h( )) represents…” -> “The random variable , with discrete probability density function h( ), represents…”

Please review all equations. I suggest that they should all be numbered and referred directly in the text, e.g. “… we include h( ) as (notation based on Shioda et al. [26]):” -> “… we include h( ) as shown in Equation X (notation based on Shioda et al. [26]).”

Page 5, line 78. “Whether or not this decrease is antigen-specific is still under debate”. Provide reference regarding this statement.

Page 6, lines 117-119. “To account for both the time to seroconversion, the time to seroreversion given seroconversion and the overall proportion that will seroconvert we include h( ) as (notation based on Shioda et al. [26]):”. In this case, ‘both’ does not fit since h(St) is included to account for three conditions, not two.

Page 6, multiple lines. “t-(t_c)”. Why is t_c in parenthesis in this case? Given t_c already has the proper underscore (in this reply block, I used the underscore to represent the lower c), is not a function and no other operation is used inside the parenthesis, could it simply be “t-t_c” instead? That alternative notation would make the equations and text a bit cleaner by removing the seemingly-unnecessary parenthesis.

Page 9, lines 212-213. “Of these persons 15% (N = 70 951) had IgG tests (N = 93 127) before July 2021.”. As I understand, first number refer to total number of persons with tests, and second number to total number of tests, but that should be clearer in the text what each number is, it is a bit confusing.

Page 13, lines 261-262. “The EuroImmun test was associated with more and faster seroreversion compared to the Wantai test.”. Why? How could the S1/RBD protein antibodies relate to the observed seroreversion differences? Is one test considered of higher quality than the other? The paper’s affirmation could be discussed.

Subsection 4.2.3. This whole section is a small paragraph with two figures. Again, better discussion over the results could be added. What could explain the behaviour of the results obtained by both tests being so different? How these align (or deviate from) with the laboratory results? How these relate to results obtained by other papers in the literature? Among other discussions.

Fig 3, page 14, line 279. “… scaled Weibull-Wi-exponential distribution…”. There is a small typo in the figure label. Please do a double-check over the entire paper for small typos like this.

Discussion section, pages 15-16. Second paragraph is enormous. Please break it down in multiple smaller paragraphs.

All figures. Figures look of low quality. Please increase the resolution/quality/DPI of the figures.

Reviewer #2: Review of Manuscript PONE-D-25-34869

The objectives of this study were to quantify SARS-CoV-2 seroconversion and reversion by time since PCR-confirmed infection, age and disease severity using data from the national/centralized database of Belgium as well as from various Laboratory based SARS-CoV-2 IgG antibody data.

The findings of this study are not novel or unique, but with the use of new approach, the findings were confirmed.

Specific comments are given below:

Laboratory data

1. It is not clear to me how the Government database or registries on PCR test results was linked with SARS-CoV-2 IgG test results done in various laboratories? Please explain/describe how the government PCR data IDs can be matched with participant IDs for antibody response with a pseudo-identifier. There is a description in lines 158 to 162, but the description is not adequate.

A section is needed to describe the LINK-VACC project, which appears to be a European registry for COVID-19 patients or vaccination. Please state what information are available from the database, e.g. age, sex, PCR testing, vaccination history etc. Whether SARS-CoV-2 IgG titers are available in database or not, how the pseudo-identifier were created and linked with individual antibody titers. This section is essential for the readers to understand and follow the design and approaches of the paper.

2. Can you please insert information about when (time line) mass vaccination started in Belgium?

3. It seems that data were sometimes were used from other European studies, not clear why this was done and whether it is justified to use these data to support the Belgian study.

Methods

• The model description is mathematically detailed — the readers may not be expected to follow the equations; simplified narrative first (with equations in supplementary materials) may improve readability and ease of understanding.

• The authors used a hierarchical Bayesian model — please provide a brief explanation about why Bayesian inference was chosen over classical alternatives (e.g., was it for better handling of uncertainty, incorporation of prior information)?

• Please give a rationale as to why “non-informative priors (normal with SD=100)” — was a reasonable choice, and whether sensitivity analyses were done with alternative priors?

• MCMC: Were convergence diagnostics besides Gelman-Rubin (e.g., trace plots, ESS) used and do they confirm robustness?

Results

• Numbers included: Could a flow diagram (e.g., exclusions step-by-step) help clarify the selection process of samples?

• Seroconversion and seroreversion: Figures are referenced, but the text could highlight key takeaways more explicitly (e.g., “By week 5, >95% had seroconverted, but durability varied strongly by test and severity”).

• Hierarchical model: Author note “effect of test/laboratories in supporting information” — summarizing the magnitude of variation in the main text may strengthen the meaning.

• Goodness of fit: Did the author provide any quantitative fit statistics in addition to plots (e.g., WAIC, DIC)?

**Do you want your identity to be public for this peer review?** For information about this choice, including consent withdrawal, please see our Privacy Policy

Reviewer #1: No

Reviewer #2: No

---

## [Author Response · Author response to Decision Letter 1]

17 Nov 2025

We would like to thank the reviewers. We uploaded a detailed 'response to reviewers' document containing the point-by-point responses.

---

## [Decision Letter · Decision Letter 1]

11 Dec 2025

Dear Dr. Braeye,

Thank you for submitting your manuscript to PLOS ONE. After careful consideration, we feel that it has merit but does not fully meet PLOS ONE’s publication criteria as it currently stands. Therefore, we invite you to submit a revised version of the manuscript that addresses the points raised during the review process.

We look forward to receiving your revised manuscript.

Kind regards,

Caio Bezerra Souto Maior, Ph.D.

Academic Editor

PLOS One

Journal Requirements:

Reviewers' comments:

Reviewer's Responses to Questions

**Comments to the Author**

Reviewer #1: All comments have been addressed

Reviewer #3: (No Response)

2. Is the manuscript technically sound, and do the data support the conclusions?

Reviewer #1: Yes

Reviewer #3: Partly

3. Has the statistical analysis been performed appropriately and rigorously?

Reviewer #1: Yes

Reviewer #3: Yes

4. Have the authors made all data underlying the findings in their manuscript fully available?

Reviewer #1: Yes

Reviewer #3: No

5. Is the manuscript presented in an intelligible fashion and written in standard English?

Reviewer #1: Yes

Reviewer #3: Yes

Reviewer #1: Additional review of some equations required - some parenthesis are bigger than others, for example in equation 2. Also, please set the equation number to be right-aligned for all equations, equation (1) for example is not right aligned

Reviewer #3: Abstract

Suggestions for improvement:

Clarity: Some sentences are long and dense. Breaking them into shorter sentences will improve readability.

Example: “Seroconversion occurred during the six weeks following a PCR-confirmed infection.” could be expanded:

“Seroconversion typically occurred within six weeks after PCR-confirmed infection. Test type, age, and disease severity strongly influenced seroconversion rates.”

Consistency of terminology: Ensure uniform use of “seroconversion” and “seroreversion” throughout.

Results section in abstract: Consider highlighting the main takeaway with a short sentence summarizing seropositivity durability by test type and severity.

Objectives

The objectives paragraph is slightly long and technical. Consider simplifying:

Methods / Model Structure

Suggestions:

Complexity / readability:

The methods are very technical. Consider providing a simpler overview first for readers less familiar with Bayesian modeling.

Keep the equations, but a brief intuitive explanation in words could help

Flow: Subheadings like “Data sources,” “Exclusion criteria,” “Bayesian model,” and “Sensitivity analysis” could improve readability.

Results

Consider summarizing key points at the beginning of the results section for readers to get the main takeaway quickly.

Some sentences are long and complex; shorter sentences would improve clarity.

Provide clear interpretation of CrIs and ORs in plain language for broader readership.

Conclusion

Could briefly mention implications for public health, e.g., improving seroprevalence studies or guiding selection of serological tests.

**Do you want your identity to be public for this peer review?** For information about this choice, including consent withdrawal, please see our Privacy Policy

Reviewer #1: No

Reviewer #3: No

---

## [Author Response · Author response to Decision Letter 2]

7 Jan 2026

Reviewer #1:

Additional review of some equations required - some parenthesis are bigger than others, for example in equation 2. Also, please set the equation number to be right-aligned for all equations, equation (1) for example is not right aligned

Thank you for spotting these formatting issues. We updated the equations.

Reviewer #3:

Thank you for taking the time again to go through this paper and improve our work.

Abstract

Suggestions for improvement: Clarity: Some sentences are long and dense. Breaking them into shorter sentences will improve readability.

Throughout the text there are several instances where we tried to decrease both the complexity and length of the sentences. Examples include the ‘objectives’ (end of introduction) and methods (abstract) section, but there were several changes throughout the paper:

Before:

"We used a hierarchical Bayesian model to estimate distributional parameters of a scaled Weibull-bi-exponential distribution for the time-varying sensitivity of qualitative serological test results obtained after PCR-confirmed SARS-CoV-2 infection."

After:

(Line 29) "We used a hierarchical Bayesian model to estimate time-varying sensitivity of serological tests following PCR-confirmed infection. The model employed a scaled Weibull-bi-exponential distribution."

Consistency of terminology: Ensure uniform use of “seroconversion” and “seroreversion” throughout.

We reviewed the document and changed instances of ‘reversion‘/’conversion’ to ‘seroreversion’/’seroconversion’.

Results section in abstract: Consider highlighting the main takeaway with a short sentence summarizing seropositivity durability by test type and severity.

We slightly rewrote the abstract to include both takeaways more clearly and to include public health implications.

(line 45) These findings highlight the need for test-specific, time-varying sensitivity adjustments in seroprevalence studies. Such adjustments are crucial for translating seroprevalence results to cumulative incidence estimates.

Objectives

The objectives paragraph is slightly long and technical.

We rewrote this part.

(line 92) We aimed to estimate time-varying, test-specific sensitivity in relation to time since infection. We also quantified the effects of age and clinical severity on this sensitivity. To estimate sensitivity, we included all data: data from Belgian laboratories and international data from the literature.

Methods

Consider simplifying: Methods / Model Structure

Suggestions: Complexity / readability: The methods are very technical. Consider providing a simpler overview first for readers less familiar with Bayesian modeling.

Keep the equations, but a brief intuitive explanation in words could help

With the paragraph directly following the ‘model structure’ heading, we aim to introduce the concepts, introducing technical/mathematical jargon, but avoiding equations/too technical parts.

(Line 167) We modeled two immunological processes: seroconversion and seroreversion. Seroconversion is the process of reaching a detectable level of antibodies after infection. Seroreversion is the process of losing that detectable level after having first attained it. We represented these as time-to-event distributions: a Weibull distribution for time to seroconversion (T_c) and a bi-exponential distribution for time to seroreversion (T_r). The Weibull distribution flexibly accommodates varying seroconversion rates. The bi-exponential distribution captures the biphasic nature of antibody decay—an initial rapid decline followed by slower waning. By combining these distributions, we calculated the proportion of individuals remaining seropositive at any given time since infection (S_t), with parameters varying by age, disease severity, and test type. We used Bayesian inference to estimate model parameters from observed test results, allowing us to quantify uncertainty in seroconversion and seroreversion estimates.

Flow: Subheadings like “Data sources,” “Exclusion criteria,” “Bayesian model,” and “Sensitivity analysis” could improve readability.

We added subheadings to the methods-section.

Results

Consider summarizing key points at the beginning of the results section for readers to get the main takeaway quickly.

We presented our key findings at the start of the discussion:

(Line 326) Our key findings demonstrate that seropositivity over time since SARS-CoV-2 infection is significantly influenced by three factors: the specific serological test used, the age of the individual, and the severity of the initial infection.

Provide clear interpretation of CrIs and ORs in plain language for broader readership.

We introduced additional sentences:

(Line 201) The Odds Ratios (OR) obtained from these coefficients quantify the relative odds of seroconversion (equation 3) or fast decay (equation 4) between groups.

(Line 230) All credible intervals (CrI, the Bayesian analogue of confidence intervals) are 95% unless otherwise stated.

Throughout the text there are several instances where we state relationships explicitly. For example:

(Line 260) Higher seroconversion was associated with older age: for 50-64 year-olds the OR was 1.29 (CrI: 1.13-1.45) and for 65-74 year-olds the OR was 1.65 (CrI: 1.41-1.97) compared to the youngest age group (18-49 years old) (Table 2).

Conclusion

Could briefly mention implications for public health, e.g., improving seroprevalence studies or guiding selection of serological tests.

We rewrote the conclusion to explicitly include these implications:

(Line 398) In conclusion, our study provided a comprehensive framework for estimating time-varying, test-specific sensitivity in serological testing for SARS-CoV-2, accounting for the critical factors of disease severity, age, and time since infection. The results demonstrate that older age (50-74 years) and more severe infection lead to higher and more durable seropositivity. Our findings have several practical implications. First, test selection substantially affects sensitivity, particularly at longer intervals post-infection; the Wantai test has higher initial sensitivity which is maintained while EuroImmun showed considerable waning of its already lower sensitivity. Second, seroprevalence estimates from populations differing in age or severity composition are not directly comparable without adjustment. Third, sensitivity estimates derived from hospitalized cohorts—common in validation studies—will overestimate sensitivity in general populations with milder infections.

---

## [Decision Letter · Decision Letter 2]

12 Jan 2026

Factors influencing SARS-CoV-2 IgG test sensitivity: A Bayesian analysis of seroconversion and seroreversion by time since infection, test, age and disease severity.

PONE-D-25-34869R2

Dear Dr. Braeye,

We’re pleased to inform you that your manuscript has been judged scientifically suitable for publication and will be formally accepted for publication once it meets all outstanding technical requirements.

Kind regards,

Caio Bezerra Souto Maior, Ph.D.

Academic Editor

PLOS One

Additional Editor Comments (optional):

Reviewers' comments:

Reviewer's Responses to Questions

**Comments to the Author**

Reviewer #3: All comments have been addressed

2. Is the manuscript technically sound, and do the data support the conclusions?

Reviewer #3: Partly

3. Has the statistical analysis been performed appropriately and rigorously?

Reviewer #3: Yes

4. Have the authors made all data underlying the findings in their manuscript fully available?

Reviewer #3: Yes

5. Is the manuscript presented in an intelligible fashion and written in standard English?

Reviewer #3: Yes

Reviewer #3: I have reviewed the manuscript thoroughly and confirm that the authors have adequately addressed all the queries and comments raised during the review process. The revisions are appropriate and clear, and I have no further queries or concerns at this time.

**Do you want your identity to be public for this peer review?** For information about this choice, including consent withdrawal, please see our Privacy Policy

Reviewer #3: No

---

## [Editor Report · Acceptance letter]

PONE-D-25-34869R2

PLOS One

Dear Dr. Braeye,

I'm pleased to inform you that your manuscript has been deemed suitable for publication in PLOS One. Congratulations! Your manuscript is now being handed over to our production team.

Kind regards,

on behalf of

Dr Caio Bezerra Souto Maior

Academic Editor

PLOS One